# Endoscopic Bilateral Optic Nerve Decompression for Treatment of Idiopathic Intracranial Hypertension

**DOI:** 10.3390/brainsci11030324

**Published:** 2021-03-04

**Authors:** Ethem Göksu, Baran Bozkurt, Deniz İlhan, Ahmet Özak, Musa Çırak, Kaan Yağmurlu

**Affiliations:** 1Department of Neurosurgery, Akdeniz University, Antalya 07070, Turkey; ahmetcan.ozak@gmail.com; 2Department of Neurosurgery, Acıbadem Mehmet Ali Aydınlar University, Istanbul 34662, Turkey; drbaranbozkurt@gmail.com; 3Department of Ophthalmology and Visual Sciences, Akdeniz University, Antalya 07070, Turkey; drdenizilhan@gmail.com; 4Department of Neurosurgery, Bakırkoy Sadi Konuk Research and Training Hospital, Istanbul 34147, Turkey; musacirak@hotmail.com; 5Department of Neurosurgery, University of Virginia, Charlottesville, VA 22903, USA; ky7zb@virginia.edu

**Keywords:** idiopathic intracranial hypertension, endoscopic optic nerve decompression, nerve sheath fenestration

## Abstract

Objective: To evaluate the results of bilateral endoscopic optic nerve decompression (EOND) with the opening nerve sheath (ONS) technique in patients with idiopathic intracranial hypertension (IIH). Methods: Between the years of 2017 and 2019, we retrospectively evaluated nine IIH patients with progressive visual impairment despite medical treatment and who were treated with the EOND and ONS techniques. We also demonstrated our surgical technique recipe on postmortem human heads in a stepwise manner. Results: There were 9 patients (7 females and 2 males) between the ages of 21 and 72 included in this study, and the mean age was 40.8. All patients had an impairment in visual acuity and/or their visual field, with signs of papilledema and/or optic atrophy. The patients were followed up with for 9–48 months. Improvements in visual acuity were observed in 7 out of 9 patients (78%). Visual field defects improved in 5 out of 8 patients (62.5%). Papilledema was resolved in all patients (100%). Headaches improved in all symptomatic patients (100%). No intraoperative or postoperative complications were observed. Conclusions: EOND is a safe and effective surgical procedure in selected patients with IIH. Bilateral wide bony decompression and nerve fenestration can also be an additional benefit for headache relief. Further clinical series and long-term follow-up are needed for more precise results.

## 1. Introduction

Idiopathic intracranial hypertension (IIH), or pseudotumor cerebri (PTC), is a clinical condition that is characterized by increased intracranial pressure (ICP) and accompanying signs and symptoms without an intracranial lesion. The associated symptoms are headache, nausea, vomiting, and visual disturbances. Papilledema is a distinctive feature of IIH, and sometimes the patients are asymptomatic with optic disc swelling, which is seen incidentally in routine eye examination [1]. Epidemiological studies have shown that this clinical syndrome predominantly affects obese women of childbearing age [2,3,4]. Radiographic features of IIH include the absence of intracranial lesions and average or small ventricular dimensions in computed tomography (CT) or magnetic resonance imaging (MRI). Additionally, empty or partially empty sella turcica, enlargement of the perioptic subarachnoid space, optic nerve tortuosity, flattening of the posterior sclera, and narrowing or stenosis of the transverse venous sinuses on vascular imaging could be observed [5,6]. Other diagnostic criteria are increased or average levels of the cerebrospinal fluid (CSF) pressure at the lumbar puncture (LP) and normal neurological examination, except diplopia [2,7].

Treatment of IIH aims to reduce ICP. Nonsurgical management is weight loss and acetazolamide or alternative drugs, including topiramate, furosemide, and octreotide. Surgical treatment modalities, including CSF diversion procedures, such as ventriculoperitoneal (VP) and lumboperitoneal (LP) shunts, optic nerve sheath fenestration (ONSF), transverse venous sinus stenting, and bariatric surgery, are considered in patients who have persisting and worsening symptoms [1,8]. Another surgical option is endoscopic optic nerve decompression (EOND) with or without opening nerve sheath (ONS), which is performed unilaterally or bilaterally by the endonasal way. To the best of our knowledge, the endoscopic bilateral optic nerve decompression technique is the least used technique in the literature. [9,10]

In this study, we evaluated the surgical outcome of 9 IIH patients who had progressive visual impairment and were treated by bilateral EOND with the ONS technique. We also demonstrated our surgical procedure in a stepwise manner on cadaveric dissections. 

## 2. Materials and Methods

Between 2017 and 2019, we retrospectively evaluated 9 IIH patients with progressive visual impairment despite medical treatment and who were treated with the EOND and ONS techniques. The diagnosis of IIH was based on modified Dandy criteria. Besides medical treatment, serial LPs were also performed in all patients to decrease ICP by draining CSF. Ophthalmological examinations and neuroradiological studies, including MRIs of the brain, orbit, and MR venography, were obtained from all patients.

### 2.1. Ophthalmological Examinations

Preoperative and postoperative ophthalmological examinations, including best-corrected visual acuity (BCVA), a fundus examination, and an automated perimetry test were recorded for all patients. The data from the automated perimetry test (Humphrey Field Analyzer II; Carl Zeiss Meditec, Dublin, CA, USA) and the cross-section images of the optic nerve head and macula from the swept-source optical coherence tomography (SS-OCT) (DRI OCT Triton Topcon Corp, Tokyo, Japan) were also evaluated.

### 2.2. Surgical Technique

All procedures were performed with the “two surgeons, four hands” technique. The head was positioned with a slight extension and left lateral flexion by a Mayfield head holder. Image guidance with a high-resolution maxillofacial CT was utilized in all cases. A 0° short rigid endoscope (4 mm, 18 cm, Hopkins II, Karl Storz, Tuttlingen, Germany) was used for the procedure. In the right nostril, the middle turbinate was pulled laterally to expose the sphenoid ostium. The anterior wall of the sphenoid sinus and posterior ethmoid air cells were drilled out and the sphenoid sinus mucosa was removed to identify the sella floor, optic prominence, carotid prominence, and opticocarotid recess as the landmarks of the sphenoid sinus. The thin, bony lamella on the intracanalicular part of the optic nerve was opened first with the help of a diamond drill, and decompression was carried out with the help of a microcurette and a 1 mm Kerrison rongeur until visualizing orbital fat tissue. After completing up to 180° of bony decompression of the optic canal (medial and inferior wall), the optic nerve sheath was incised with a micro knife crescent to visualize arachnoid herniation and minimal CSF drainage. The same procedure was repeated on another side (Appendix A) (Figure 1). During the closure, the sinus was not packed with any material.

### 2.3. Cadaveric Demonstration 

We demonstrated our surgical technique on postmortem human heads, which do not have any sinonasal or intracranial pathology, in a stepwise manner. We prepared 3 human head specimens for dissection at the Surgical Neuroanatomy Laboratory of the Department of Neurological Surgery at the University of Virginia, USA. Endoscopic endonasal anatomic dissections were performed using rod lens endoscopes (4 mm, 18 cm, Hopkins II, 0° and 45°, Karl Storz, Tuttlingen, Germany). Anatomical dissections were documented with a high-resolution camera and a digital video recording system. The heads were positioned in a supine and fixed neutral position by a Mayfield head holder on a workstation. The optic canal and the opticocarotid recess were demonstrated through an endoscopic endonasal approach to the sella and the tuberculum sella region (Figure 2).

## 3. Results

There were 9 patients (7 females and 2 males) between the ages of 21 and 72 included in this study, and the mean age was 40.8. MR venography examinations were normal in all patients. Two patients were pregnant (Cases 6 and 7). All patients had an impairment in visual acuity and/or visual field. There were also signs of papilledema and/or optic atrophy with subretinal fluid, retinal edema, and hemorrhages on the optic disc to different degrees in all patients. Headaches were observed in 6 out of 9 patients as another presenting symptom. Table 1 demonstrates the demographic and preoperative clinical features of all patients. An LP shunt procedure was performed in one patient (Case 2) at another center, a partial improvement was achieved, but visual impairment was persistent. In this case, despite the severe visual findings and bilateral optic atrophy, surgery was planned according to the visual evoked potential (VEP) test. Postoperatively, a little improvement in visual acuity was achieved, and the color vision has started on the left eye, but the VEP response was stable. The patients were followed up with for 9–48 months. Improvements in visual acuity were observed in 7 out of 9 patients (78%). Visual field defects improved in 5 out of 8 patients (62.5%). The preoperative and postoperative visual field examinations are demonstrated in Figure 3. Papilledema was resolved in all patients (100%). Headaches improved in all symptomatic patients (100%). Table 2 illustrates the postoperative clinical features of the patients. The preoperative and postoperative visual findings are demonstrated in three illustrative cases (Cases 7–9) in Figure 4, Figure 5 and Figure 6, respectively. No intraoperative or postoperative complications, such as a CSF leak, meningitis, and so on, were observed. 

## 4. Discussion

IIH or PTC is a clinical presentation of an uncertain etiology. It is characterized by increased intracranial pressure and associated symptoms and signs such as headaches, nausea, pulsatile tinnitus, visual loss, visual field defects, and papilledema. It is mostly seen in childbearing women, and the incidence increases up to 11.9/100.000 with obesity [11]. Progressive papilledema, visual impairment, or intractable headaches are the most common clinical indications for surgical treatment.

CSF diversion procedures are still the most frequently used among surgical therapies. Following CSF shunt procedures, a 96% improvement in headaches, a 59.9–85.7% improvement in papilledema, a 49.3–56.6% improvement in visual acuity, and a 68.4–78.3% improvement in the visual field were reported [12,13]. However, LP shunt procedures’ significant problems are high revision rates due to obstruction, malposition, or overdrainage. In VP shunt procedures, catheter placement can also be a technical challenge because of slit ventricles. On the other hand, the progression of visual loss was also reported after the shunt procedure in some cases. 

ONSF is another surgical option in patients with papilledema and progressive vision loss despite medical therapy. Ophthalmic surgeons have performed the procedure for many years in different approaches, including medial, lateral, superior, or combined methods. It is also known to improve bilateral vision loss and severe papillary edema, even if performed on one side [14]. In a comprehensive review by Gilbert et al., improved papilledema, visual acuity, and visual field defects were reported as 95%, 67%, and 64%, respectively. The authors also stated headache relief in 41% of patients. However, procedure-related complications, including oculomotor, abducens nerve palsies, orbital hematoma, pupillary changes, and orbital apex syndrome, were reported almost 30% [1]. Furthermore, the procedure is always performed unilaterally, and bilateral fenestration is not possible in one stage.

EOND is a new surgical option in patients who have predominantly visual findings with IIH. In 2007, Gupta et al. presented the results of 18 IIH patients treated with unilateral EOND with ONS. They reported the visual improvement of 17 out of 18 patients on the ipsilateral side and 12 out of 15 patients on the contralateral side [4]. In 2014, Sencer et al. reported 10 consecutive cases of unilateral EOND without ONS. According to their findings, the visual field defects and visual acuity improved in 8 out of 9 patients, whereas papilledema improved in 7 out of 9 patients. Furthermore, headaches resolved in 4 out of 7 patients. However, contralateral eye findings were similar in half of the patients, and no changes were observed [15]. In 2017, Tarrats et al. reported a systematic review that examined the outcomes of EOND in patients with IIH. This review included six studies with a total of 34 patients. Although the studies were not homogeneous (unilateral/bilateral EOND with/without ONS), overall results were significant and improved visual field deficits, visual acuity, papilledema, and headaches were reported as 93.8%, 85.3%, 81.4%, and 81.8%, respectively [9]. There were also no significant complications reported with this approach. Recently, Srivastava et al. presented their experiences with 9 patients with idiopathic and secondary intracranial hypertension due to cerebral venous sinus thrombosis. In their study, all patients underwent bilateral EOND with ONS, and they reported the improvement in visual acuity and headaches in 6 out of 9 patients. In contrast, the visual acuity deteriorated in 2 patients [10]. In our study group, we found the improvement rates of visual acuity, visual field defect, papilledema, and headaches were 78%, 62.5%, 100%, and 100%, respectively, without any operative complications. We observed the improvements in some visual parameters, even in patients who had optic atrophy preoperatively (Cases 1–6). The prognosis was not favorable in these delayed cases, as expected. However, the patients who had progressive visual deterioration and underwent early EOND with ONS showed an excellent prognosis (Cases 7–9).

A comprehensive literature review compared the outcomes of the surgical treatment modalities for intracranial hypertension. It was noted that CSF diversion techniques diminished papilledema, visual field deterioration, and headaches in 79%, 67%, and 70% of the cases, respectively, and were associated with a 9% severe complication rate and a 43% failure rate. In contrast, ONS fenestration ameliorated papilledema, visual field defects, and headaches in 90%, 65%, and 49% of patients, respectively, and the severe complication rate was 2%, and the failure rate was 9% [16].

The type of surgery, unilateral or bilateral, with or without ONS, is the main difference among the several studies. Some authors recommended the ONS to decrease the CSF pressure, which is transmitted to optic nerves. They suggested that ONS was necessary to improve the visual symptoms, and only bony decompression would not decrease the pressure that comes from within the nerve [17,18,19]. Improvement in contralateral eye findings with unilateral ONS supports the importance of nerve fenestration [1,17]. Another advantage of ONS is headache relief. In a review of Tarats et al., headache relief was reported to be significantly higher in patients who underwent EOND with ONS than in patients who underwent only EOND [9]. Gilbert et al. also stated the 41% of headache relief in a review of unilateral transorbital ONS [1]. Our findings were consistent with the literature on the effect of sheath fenestration on headache relief. We observed the improvement in headaches in all symptomatic patients. Headache is the most common symptom in IIH patients, and it seriously affects the quality of life. Therefore, the headache should not be ignored during surgery planning for IIH cases. A possible mechanism of the ONS effect on headache relief could be explained by the decrease in ICP and meningeal tension by CSF release. In our study group, we used bilateral EOND with ONS in all patients. We think that a wide bilateral opening of the optic canals from chiasm to orbital apex and ONS could improve visual disturbances and headaches. Technically, the endoscopic midline transsphenoidal way is a familiar approach for neurosurgeons. Bilateral bony decompression of the optic canals by at least 180° and ONS are not time-consuming or complicated processes. The main advantage of this approach is to decompress the bilateral optic nerves in one stage with minimal morbidity. To our knowledge, our series is the most comprehensive group, including the IIH patients who underwent bilateral EOND with ONS. However, the major limitation of this study is the heterogeneous patient population. Some patients had severe optic atrophy preoperatively, and they demonstrated a stable course due to delayed clinical situations. In this group, the procedure might be beneficial to prevent clinical progression and provide headache relief.

## 5. Conclusions

EOND is a safe and effective surgical procedure in selected patients with IIH. Early surgery seems to be highly efficient in improving visual findings in patients who have progressive deterioration and no optic atrophy. Bilateral wide bony decompression and nerve fenestration can also be an additional benefit for headache relief. Further clinical series and long-term follow-ups are needed for more precise results.

## Figures and Tables

**Figure 1 brainsci-11-00324-f001:**
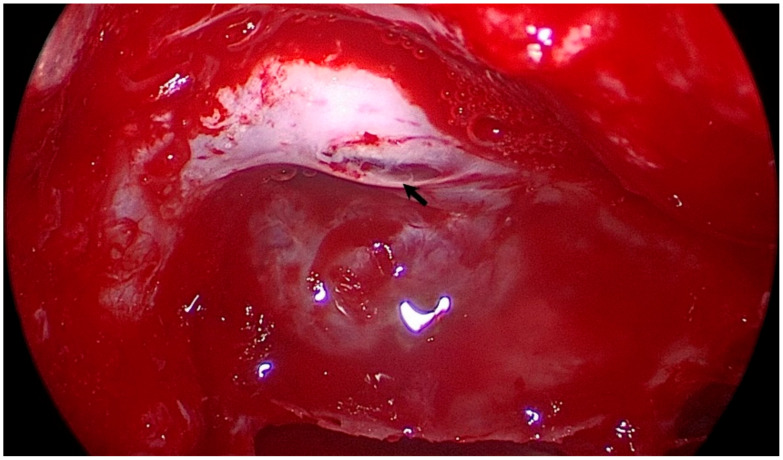
Intraoperative demonstration of right optic nerve following bony decompression and opening nerve sheath (black arrow).

**Figure 2 brainsci-11-00324-f002:**
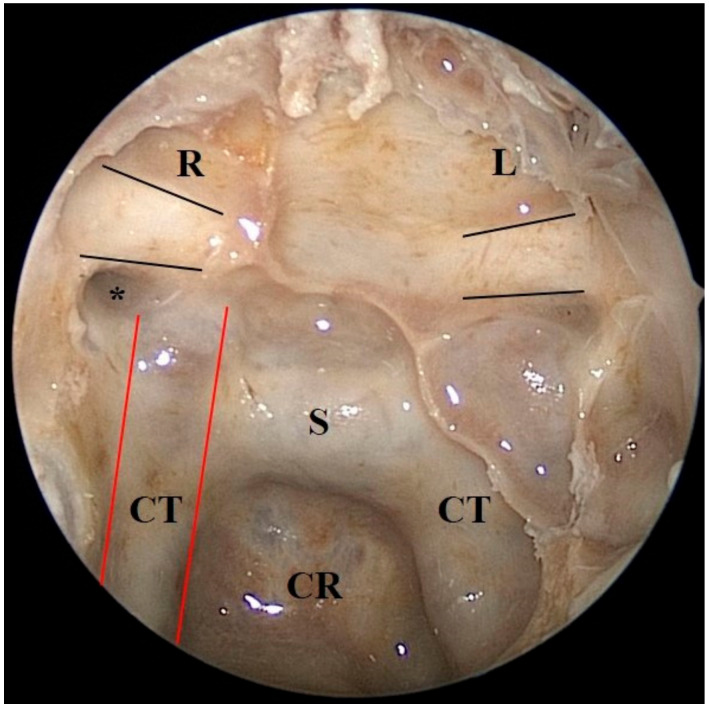
Cadaveric demonstration of anatomical landmarks in the sphenoid sinus. The two black lines outline the canalicular segment of the optic nerve. The two red lines show the course of the internal carotid artery in the sphenoid sinus. * points to the opticocarotid recess. CR: Clival Recess; CT: Carotid Tubercule; S: Sella.

**Figure 3 brainsci-11-00324-f003:**
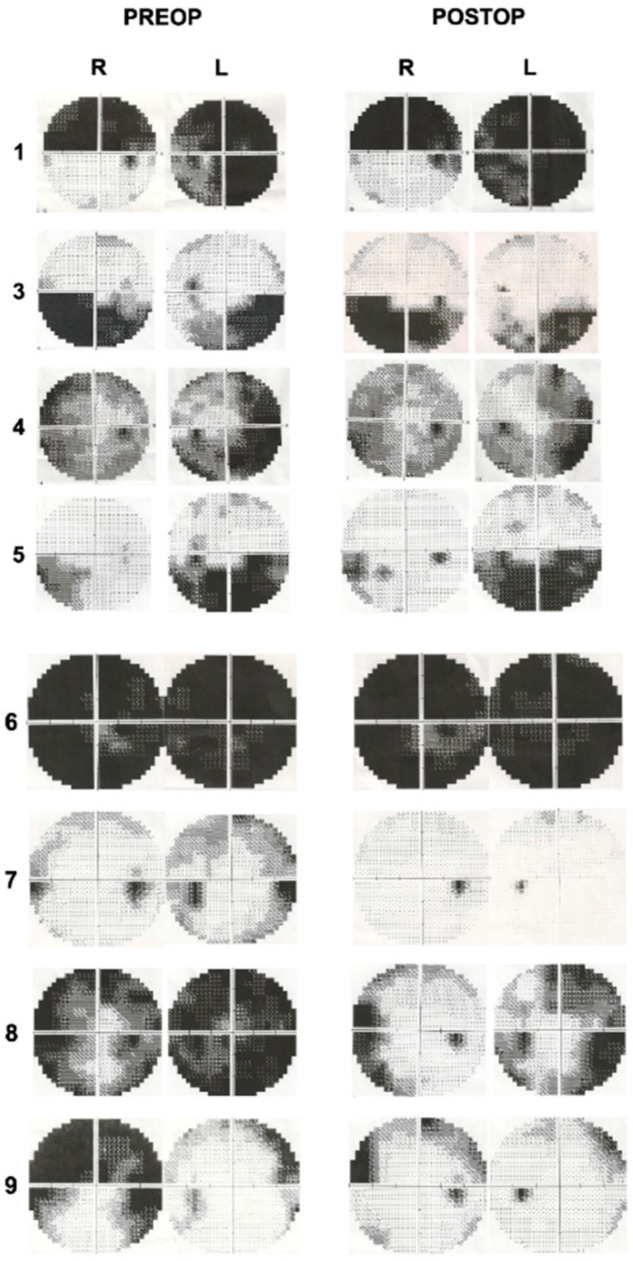
Preoperative and postoperative visual field examinations of patients.

**Figure 4 brainsci-11-00324-f004:**
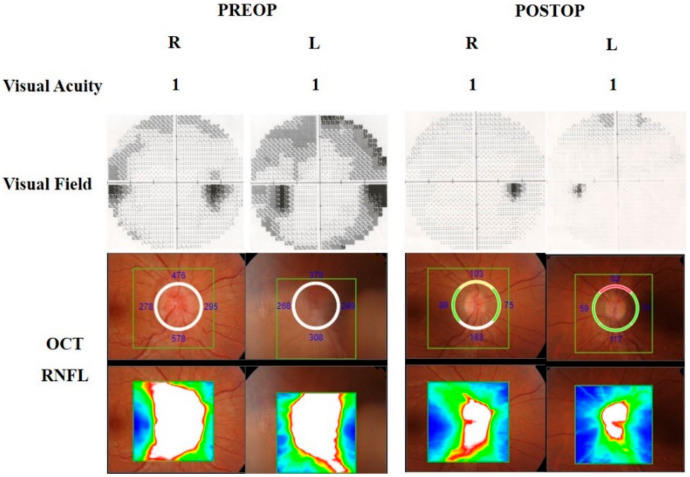
The 32-year-old pregnant patient (Case 7) was consulted with headaches and progressive visual impairment. The visual field showed bilateral enlargement of a blind spot with a general reduction of sensitivity; peripheral nasal defects in the right eye and peripheral constriction apparently in the upper hemisphere in the left eye. Fundus examination revealed severe optic disc swelling. Retinal nerve fiber layer (RNFL) in optical coherence tomography (OCT) showed bilateral increased retinal nerve fiber thickness due to papilledema. Three months following endoscopic optic nerve decompression (EOND), bilateral RNFL thickness was decreased, and visual field defects improved.

**Figure 5 brainsci-11-00324-f005:**
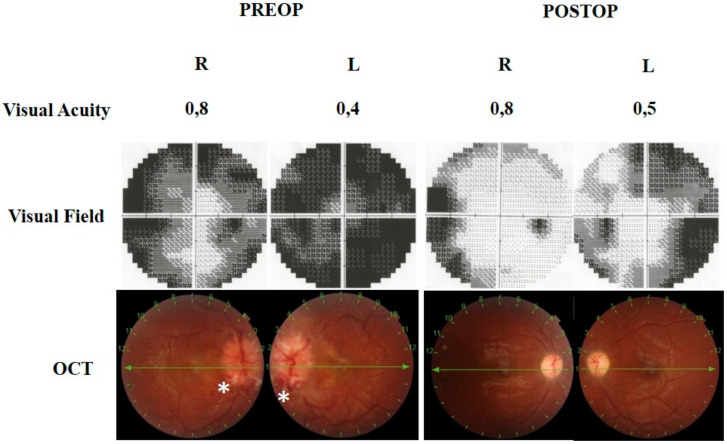
The 21-year-old male patient (Case 8) presented with headaches and progressive visual impairment. The visual field showed severe bilateral constriction of visual field defect with a general reduction in sensitivity. There was also severe bilateral papilledema with optic disc hemorrhages (*) and retinal vein engorgements, choroidal folds, and retinal edema. One month following EOND, the visual field defects improved, retinal hemorrhages disappeared, and edema noticeably resolved.

**Figure 6 brainsci-11-00324-f006:**
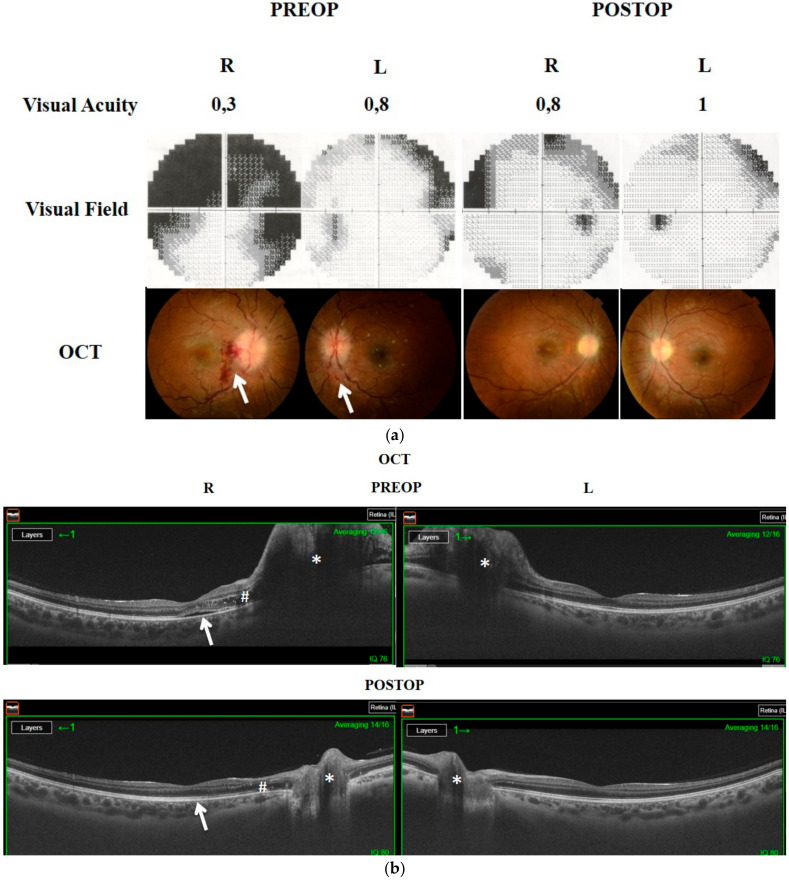
(**a**). The 28-year-old female (Case 9) presented with headaches and progressive visual loss, particularly in the right eye. Visual acuity was 0.3/0.8, and the visual field revealed a large defect in the right eye. She had a severe degree of optic disc edema, markedly vascular engorgement, and retinal folds with bleeding in the temporal peripapillary region. White arrows show the hemorrhages in the right and left eye. Two months following EOND, visual acuity, the visual field, and OCT findings improved. (**b**). Preoperative OCT also demonstrated bilateral optic disc swelling (*) with retinal edema (#) and subretinal fluid under fovea (white arrow) in the right eye. Postoperatively, the bilateral optic disc swelling noticeably improved (*), retinal edema (#), and subretinal fluid (arrow) regressed in the right eye.

**Table 1 brainsci-11-00324-t001:** Demographic and preoperative clinical features of patients.

Patient	Age	Sex	BMI	Headache	Visual Acuity (BCVA) R/L	Visual Field Defect	Fundoscopic Examination (Papilledema/OA)
**1**	72	Male	>30	−	0.4/0.2	+	−/+
**2**	54	Female	>30	+	LP/FC	NA	−/+
**3**	37	Female	>30	−	0.5/0.6	+	+/+
**4**	54	Female	>30	−	0.7/1	+	−/+
**5**	47	Female	>30	+	0.9/0.9	+	+/+
**6**	23	Female	>30	+	0.6/0.6	+	+/+
**7**	32	Female	<30	+	1/1	+	+/−
**8**	21	Male	<30	+	0.8/0.4	+	+/−
**9**	28	Female	>30	+	0.3/0.8	+	+/−

BCVA: Best Corrected Visual Acuity; BMI: Body Mass Index; LP: Light Perception; FC: Finger Counting; OA: Optic Atrophy.

**Table 2 brainsci-11-00324-t002:** Postoperative clinical features of patients. BCVA: Best Corrected Visual Acuity.

Patient	Headache	Visual Acuity (BCVA)	Visual Field Defect	Fundoscopic Examination
		R/L		Papilledema/OA
**1**	−	0.5/0.2	Unchanged	Unchanged
**2**	Improved	LP/0.05	NA	Unchanged
**3**	−	0.7/0.8	Unchanged	Resolved/+
**4**	−	0.7/1	Improved	Unchanged
**5**	Improved	1/0.9	Improved	Resolved/+
**6**	Improved	0.9/0.9	Unchanged	Resolved/+
**7**	Improved	1/1	Improved	Resolved/−
**8**	Improved	0.8/0.5	Improved	Resolved/−
**9**	Improved	0.8/1	Improved	Resolved/−

LP: Light Perception; OA: Optic Atrophy.

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
