# Peer review of "Endoscopic Bilateral Optic Nerve Decompression for Treatment of Idiopathic Intracranial Hypertension"

_brainsci, 2021, doi:10.3390/brainsci11030324_

Round 1

Reviewer 1 Report

First of all, congratulations on your work! the subject is very interesting and actual, and the paper is well-documented. Although the number of cases is small and the cases are heterogeneous, the research is a useful contribution in describing the outcomes of the described technique (EOND with ONS).

Minor point:

The information in the second paragraph in the Introduction section and the first paragraph of the Discussion section is somehow redundant. I suggest that this information regarding alternative treatments for IIH could be comprised in the Introduction, while in the Discussion section could remain only comparative information on the technique described in the paper. 

Author Response

25th Feb 2021

Ms. Roxana Hutanu

Assistant Editor

MDPI Open Acces Publishing Romania SRL

Dear Editor,

Thank you for your e-mail communicating the overall view of our manuscript and a request for addressing specific issues in a revised resubmission. On behalf of all my co-authors, I would also like to thank the editorial team and reviewers for a comprehensive and insightful review of our submitted manuscript. The comments of the reviewers have helped us identify areas for improvement. We have included an improved table and text changes in the revised manuscript. Specific replies to reviewer’s comments are detailed below and are highlighted in the revised text. We have attempted to address the reviewer comments to the best of our abilities and we hope you would favorably consider the revised manuscript for publication in your journal. Thank you for your kind attention.

On behalf of all co-authors,

Ethem GÖKSU, MD.

Reviewer 1.

  • First of all, congratulations on your work! the subject is very interesting and actual, and the paper is well-documented. Although the number of cases is small and the cases are heterogeneous, the research is a useful contribution in describing the outcomes of the described technique (EOND with ONS).
  • Minor point:

The information in the second paragraph in the Introduction section and the first paragraph of the Discussion section is somehow redundant. I suggest that this information regarding alternative treatments for IIH could be comprised in the Introduction, while in the Discussion section could remain only comparative information on the technique described in the paper. 

#Response:We thank the reviewer for calling this point to our attention. As you suggested, we removed the fourth sentence of first paragraph in discussion section.

Reviewer 2 Report

The authors retrospectively reviewed a single author series of bilateral endoscopic optic nerve decompression (EOND) with opening nerve sheath (ONS) technique in nine patients harboring idiopathic intracranial hypertension (IIH).  Improvements in visual acuity were observed in seven of nine patients (78%), whereas papilledema was resolved in all patients, without intraoperative and postoperative complications

The topic is really intriguing, and the current work could add relevant information about the management of such intriguing diseases. Also, the figures clearly depict that main information. However, I have some issues that must be clarified.

It should be interesting to know if the authors compared the results of the EOND with a historical institutional (or single author) series, for example, of patients treated with shunt procedures. Moreover, authors should better clarify why they excluded some "more traditional" treatment (namely diversion procedures)

No data about the intracranial pressure as measured during the lumbar punctures? I also think that patients' weight should be added.

Patients 1: a 72 years old male is not exactly the typical patient with IIH, and also his clinical presentation is quite strange (no headache, and unchanged visual defect after surgery). 

Author Response

25th Feb 2021

Ms. Roxana Hutanu

Assistant Editor

MDPI Open Acces Publishing Romania SRL

Dear Editor,

Thank you for your e-mail communicating the overall view of our manuscript and a request for addressing specific issues in a revised resubmission. On behalf of all my co-authors, I would also like to thank the editorial team and reviewers for a comprehensive and insightful review of our submitted manuscript. The comments of the reviewers have helped us identify areas for improvement. We have included an improved table and text changes in the revised manuscript. Specific replies to reviewer’s comments are detailed below and are highlighted in the revised text. We have attempted to address the reviewer comments to the best of our abilities and we hope you would favorably consider the revised manuscript for publication in your journal. Thank you for your kind attention.

On behalf of all co-authors,

Ethem GÖKSU, MD.

Reviewer 2.

  • The authors retrospectively reviewed a single author series of bilateral endoscopic optic nerve decompression (EOND) with opening nerve sheath (ONS) technique in nine patients harboring idiopathic intracranial hypertension (IIH).  Improvements in visual acuity were observed in seven of nine patients (78%), whereas papilledema was resolved in all patients, without intraoperative and postoperative complications. The topic is really intriguing, and the current work could add relevant information about the management of such intriguing diseases. Also, the figures clearly depict that main information. However, I have some issues that must be clarified.
  • It should be interesting to know if the authors compared the results of the EOND with a historical institutional (or single author) series, for example, of patients treated with shunt procedures. Moreover, authors should better clarify why they excluded some "more traditional" treatment (namely diversion procedures).

#Response:

We thank the reviewer for calling this oversight to our attention. Since we do not have case series treated with CSF diversion procedures, we have recently added a comprehensive literature review that compared outcomes of the CSF diversion techniques and ONS fenestration to the discussion section as follows: “A comprehensive literature review compared the outcomes of the surgical treatment modalities for the intracranial hypertension. It was noted that CSF diversion techniques diminished papilledema, visual field deterioration, and headache in 79%, 67%, and 70% of the cases and are associated with a 9% severe complication rate and a 43% failure rate. In contrast, ONS fenestration ameliorated papilledema, visual field defects, headaches in 90%, 65%, and 49% of patients, and the severe complication rate was 2%, and the failure rate was 9%.“

      Newly added citation:

Kalyvas A, Neromyliotis E, Skandalakis GS, Koutsarnakis C, Komaitis S, Zadeh G, Gentili F, Gobin PY, Stranjalis G, Patsalides A. A Systematic Review of Surgical Treatments of Idiopathic Intracranial Hypertension (IIH): Should VSS Be Regarded as the First Line Surgical Modality ?. Neurosurgery 2020;67(Supplement 1):447-315.

  • No data about the intracranial pressure as measured during the lumbar punctures? I also think that patients' weight should be added.

#Response: We thank the reviewer for calling this point to our attention. In all patients, intracranial pressures as measured during the LP’s were over 25 cmH2O. In the first paragraph of materials and methods section, we already mentioned that “The diagnosis of IIH was based on modified Dandy criteria”.  We also added patients’ BMI information as an extra column in table 1.

Patient

Age

Sex

BMI

Headache

Visual Acuity (BCVA)

R/L

Visual Field Defect

Fundoscopic Examination

(Papilledema/OA)

1

72

Male

> 30

-

0,4/0,2

+

-/+

2

54

Female

> 30

+

LP/FC

NA

-/+

3

37

Female

> 30

-

0,5/0,6

+

+/+

4

54

Female

> 30

-

0,7/1

+

-/+

5

47

Female

> 30

+

0,9/0,9

+

+/+

6

23

Female

> 30

+

0,6/0,6

+

+/+

7

32

Female

< 30

+

1/1

+

+/-

8

21

Male

< 30

+

0,8/0,4

+

+/-

9

28

Female

> 30

+

0,3/0,8

+

+/-

      Table 1.Demographic and preoperative clinical features of patients. BCVA; Best Corrected Visual Acuity, BMI; Body Mass Index, LP; Light Perception, FC; Finger Counting, OA; Optic Atrophy.

  • Patients 1: a 72 years old male is not exactly the typical patient with IIH, and also his clinical presentation is quite strange (no headache, and unchanged visual defect after surgery). 

#Response: Thank you for calling this point. This 72-year-old male patient presented with progressive visual loss. Intracranial pressure as measured during the LP’s was over 25 cmH2O. There was no other reason detected by ophthalmologist. Despite the acetazolamide treatment, visual field findings deteriorated and we decided to perform EOND even if he was not a typical patient. Postoperatively, the patientdemonstrated a stable course and the procedure might be beneficialto prevent clinical progression.We pointed in the last sentence of the discussion section as“..In this group, the procedure might be beneficialto prevent clinical progression and headache relief”.

Round 2

Reviewer 2 Report

The authors answered to my questions